# Fast Phylogeny of SARS-CoV-2 by Compression

**DOI:** 10.3390/e24040439

**Published:** 2022-03-22

**Authors:** Rudi L. Cilibrasi, Paul M. B. Vitányi

**Affiliations:** 1Centre for Nathematics & Computer Science CWI, Science Park 123, 1098 XG Amsterdam, The Netherlands; cilibrar@gmail.com; 2CWI (Centrum Wiskunde & Informatica), Department of Computer Science, Faculteit Natuurwetenschappen, Wiskunde en Informatica, University of Amsterdam, Science Park 904, 1098 XH Amsterdam, The Netherlands

**Keywords:** compression, phylogeny, COVID-19 virus

## Abstract

The compression method to assess similarity, in the sense of having a small normalized compression distance (NCD), was developed based on algorithmic information theory to quantify the similarity in files ranging from words and languages to genomes and music pieces. It has been validated on objects from different domains always using essentially the same software. We analyze the whole-genome phylogeny and taxonomy of the SARS-CoV-2 virus, which is responsible for causing the COVID-19 disease, using the alignment-free compression method to assess similarity. We compare the SARS-CoV-2 virus with a database of over 6500 viruses. The results suggest that the SARS-CoV-2 virus is closest in that database to the RaTG13 virus and rather close to the bat SARS-like coronaviruses bat-SL-CoVZXC21 and bat-SL-CoVZC45. Over 6500 viruses are identified (given by their registration code) with larger NCDs. The NCDs are compared with the NCDs between the mtDNA of familiar species. We address the question of whether pangolins are involved in the SARS-CoV-2 virus. The compression method is simpler and possibly faster than any other whole-genome method, which makes it the ideal tool to explore phylogeny. Here, we use it for the complex case of determining this similarity between the COVID-19 virus, SARS-CoV-2 and many other viruses. The resulting phylogeny and taxonomy closely resemble earlier results from by alignment-based methods and a machine-learning method, providing the most compelling evidence to date for the compression method, showing that one can achieve equivalent results both simply and quickly.

## 1. Introduction

In the 2019–2022 pandemic of the COVID-19 disease caused by the SARS-CoV-2 virus [1], studies of the phylogeny and taxonomy of this virus use essentially two methods: alignment-based analysis, for example [2]; and alignment-free machine learning [3].

In alignment-based methods, one customarily takes two or more sequences over a bounded alphabet of letters and makes them identical by changing letters, introducing gaps, etc., where each of these operations carries a cost. The total cost is computed by adding these sub costs. The alignment with the least total cost is preferable. Problems include, e.g., similar regions on sequences being compared may be far apart, causing massive alignment costs and a computation cost for an alignment which may be too high or forbidding. Alignment methods have many more drawbacks which make alignment-free methods more attractive [4,5,6,7].

Machine learning algorithms learn a concept using an input consisting of many examples of that concept. It is alignment-free but uses complicated algorithms with many parameters to set. This was used in the phylogeny analysis of the COVID-19 virus in [3] together with decision trees and other devices.

The purpose of this study was to identify the closest viruses to the SARS-CoV-2 virus isolate and determine their structural relations (taxonomy) in phylogeny trees in Figure 1 and Figure 2 and the NCDs in Table 1 using the compression method. This method is alignment-free, based on the lossless compression of the whole-genome sequences of base pairs [8] of the involved viruses, and is called the normalized compression distance (NCD) method or *compression method* for short. The method computes the NCDs between pairs of genomes resulting in an NCD-matrix.

We computed the NCDs between 6751 viruses and a selected SARS-CoV-2 virus but we can only visualize part of them. With this method, we quantify the proximity relations between pairs of viruses by their NCDs and compare them to similar relations between the mtDNAs of familiar mammal species in order to gain an intuition as to what they mean.

Phylogeny analysis is widely used but there are challenges of, among others, accuracy and speed. As a measure of similarity, the NCD seems a good idea to solve these challenges. The compression method determines the similarity of two sequences using a single formula once, and thus does not use many examples. Since the NCD assesses similarity for sequences in all domains, we apply it to the complex problem of SARS-CoV-2 phylogeny. We verify whether the results using this method are comparable with those from the other used methods. The conclusion is that the NCD method is an alignment-free method that gives accurate phylogeny results, while being possibly faster than both alignment-based methods and other alignment-free methods.

Previous studies have pointed to bats as the origin of the SARS-CoV-2 virus. It is thought to belong to lineage B (Sarbecovirus) of Betacoronavirus. From phylogenetic analysis and genome organization, it was identified as a SARS-like coronavirus, and to have the highest similarity to the SARS bat coronavirus RaTG13 and similar to the two bat SARS-like coronaviruses bat-SL-CoVZXC21 and bat-SL-CoVZC45, see for example [2,3]. We computed the NCDs of 6751 + 15,409 = 22,160 virus pairs plus the NCDs used for Figure 1 and Figure 2. Altogether, this took approximately 5–10 h in a combined run on a home desktop computer (a mini-computer called Meerkat from a Linux computer company called System76). It uses less than 2 s per pair which comes to more than 2000 pairs per hour. The same program can be re-used for different phylogenies and questions. This is possibly the easiest and fastest method to establish whole-genome phylogeny.

Figure 1 and Figure 2 were built with the Phylogeny Inference Package (PHYLIP) [9] and Appendix A with the CompLearn Package [10]. The programming to obtain the NCD matrices underlying the phylogenetic trees in the figures is simple; essentially, Equation (Equation 1) is computed for all pairs of viruses to be compared. The comparisons between the NCDs of viruses and the NCDs of the mtDNA of species are made on the basis of Appendix A or [11].

## 2. Method

A *string* is a sequence of finite length over a finite alphabet—here A, C, G, T. Let *Z* be a lossless compressor (‘lossless’ means that the original can be reconstructed from the compressed version, that is, nothing of the compressed string gets lost). Use Z(z) with z=x,y,xy, where xy is the concatenation of the *x* and *y* (file *y* appended to file *x*), to denote the length of the compressed version of string *z*. The *normalized compression distance* (NCD) is given by
(1)$$NCD_{Z}\left(x,y\right)=\frac{Z(xy)-\min(Z(x),Z(y))}{\max(Z(x),Z(y))},$$
where we eliminate the subscript ‘*Z*’ when the compressor *Z* is understood. A complete mathematical derivation of this distance appeared in [12,13] and especially in [11] where the name NCD was coined. A brief derivation is given in Section 7.

An explanation using a lot of hand-waving is as follows. The compressed version of xy with length Z(xy) can be viewed as the concatenation of the compressed version of string *x* with the help of string *y* and the compressed version of string *y* with the help of string *x*. Subtracting Z(x) from Z(xy) yields at most the length of the compressed version of *y* with the help of *x*, which we may write as Z(y|x), while subtracting Z(y) from Z(xy) yields at most Z(x|y). Assume that *Z* compresses every string to the shortest string from which it can be reconstructed. Then, we can replace “at most” with “about”. The numerator Z(xy)−min(Z(x),Z(y)) consists of the maximum of the two lengths Z(x|y) and Z(y|x). This in turn can be viewed as a distance expressing resemblance. For example, if *x* is a string and *y* is the empty string (the string of length 0 without any letters), then this distance is Z(x). Since string *x* may be larger than string *y* or vice versa, if we want to determine the similarity of *x* and *y*, then we need to normalize. The denominator max(Z(x),Z(y)) is the normalizing factor taking care that the NCD is between 0 and 1 in all cases.

The NCD is a family of compression functions parameterized by the used data compressor *Z*. In a compressor, the “window size” is crucial. Window size is an upper bound on the number of bytes of the input: here, it is essentially the summed number of bytes in the two objects *x* and *y* being compared. Common compressors are gzip, which is a Lempel–Ziv compressor with a small sliding window size of 32 kB, bzip2, which is a block-sorting compressor based on the Burrows–Wheeler transform with a larger window size of 256 kB, and PPMZ, which uses an adaptive statistical data compression technique based on context modeling and prediction and with unlimited window size [14]. The objects being compared for similarity must fit in approximately one-half of the window size.

In [11], there are given necessary and sufficient conditions for computing the NCD with a compression program *Z*. The NCD is an approximation of a concept called NID with the same Formula (Equation 1) with *Z* replaced by the Kolmogorov complexity *K* (for Kolmogorov complexity, see Section 7). Since K(x) is close to a greatest lower bound on Z(x), and $NCD_{K}\left(x,x\right)=0$, the $NCD_{Z}$ turns out to be more accurate the better the compressor *Z* satisfies $NCD_{Z}\left(x,x\right)=0$ and this also holds for other properties (symmetry, identity, resistance to noise). These properties are to various degrees satisfied by good real-world compressors such as gzip, bzip2, PPMZ, and zpaq. In [15,16], the authors systematically investigated how far the performances of compressors gzip, bzip2, and PPMZ satisfy these properties. With Z= gzip usually $NCD_{Z}\left(x,x\right)$ is between $\frac{1}{2}$ and 1 (very bad). With Z= bzip2, it is lower but nowhere near 0, and $NCD_{PPMZ}\left(x,x\right)$ usually is, for example in the genomic experiment of Appendix A, in between 0.002 and 0.006.

We used zpaq as an compression algorithm since [17] “bench marked 430 settings of 48 representative compressors (including 29 specialized sequence compressors and 19 general-purpose compressors) on 27 representative FASTA-formatted test data sets including individual genomes of various sizes, DNA and RNA data sets, and standard protein data sets”. Figure 1A of the cited paper compares the compressors with respect to their compression ratio: zpaq is the best of the considered general-purpose compressors and competitive with the best considered specialized sequence compressors.We computed the $NCD_{zpac}\left(x,x\right)$ with *x*, the selected SARS-CoV-2 virus isolate serving as standard for the NCDs with other viruses, as can be seen in Section 3, in two cases:The first 1700 bytes of the selected sequence where the compressed size alone is 445 and paired with itself (concatenated) is 460, yielding an NCD of 0.0337078651685393.For the full virus sequence, (approximately 28,000 bases) the compressed size alone is 7181 and when paired with itself, it has a compressed size of 7208, yielding an NCD of 0.00375992201643225.

The second item shows that, in this respect, zpac is similar to PPMZ (the mtDNA sequences are typically 17,000 bases), while the two items together show that the $NCD_{zpac}$ is not linear.

### 2.1. Validation of the Method

Researchers from the data-mining community noticed that the NCD is a parameter-free, feature-free, data-mining tool. They have tested a simplified version of the NCD on a large variety of sequence benchmarks. Comparing the compression method with 51 major parameter-loaded methods found in the seven major data-mining conferences (SIGKDD, SIGMOD, ICDM, ICDE, SSDB, VLDB, PKDD, and PAKDD) in a decade, on every database of time sequences used, ranging from heartbeat signals to stock market curves, they established the clear superiority of the compression-based method for clustering heterogeneous data, for anomaly detection, and competitiveness in the clustering domain data [18]. Many researchers use this method, introduced in [11,13], for example [19,20,21,22]. In the Appendix A, we treat two examples from [11], testing this method on known phylogenies before applying it to complex novel data such as those of the SARS-CoV-2 virus.

### 2.2. How We Used the Method

Using the compression method to compute the phylogeny of the SARS-CoV-2 virus, we first obtained a list of over 6500 viruses that we wanted to compare to the SARS-CoV-2 virus, without duplicates, including partially sequenced viruses, and SARS-CoV-2 viruses (all apart from one). A particular SARS-CoV-2 virus isolate served as the standard against which to determine the NCDs of the other viruses. We selected this SARS-CoV-2 virus isolate from the many, at least 15,500, examples available on 17 July 2020 in the GISAID database. We computed for each virus in the list of over 6500 viruses its NCD with the selected SARS-CoV-2 virus isolate. As the compression program, we used zpaq. Subsequently, we ordered the resulting NCDs from the smallest to the largest. The virus causing the smallest NCD distance with the SARS-CoV-2 virus isolate was regarded as the most similar to that virus. We took the 60 viruses which had the least NCDs with the selected SARS-CoV-2 virus isolate and computed the phylogeny of those viruses and the SARS-CoV-2 virus isolate. We then compared the last mentioned virus with 37 close viruses including pangolin ones, to determine the relation between the SARS-CoV-2 virus and the pangolin viruses.

## 3. Materials

We downloaded the data in two parts and stored them on the public repository GitHub. The two data sets were as follows. One data set was obtained from the authors of the machine-learning-approach study [3] and consisted of the 6751 virus sequences used in that paper. There was one SARS-CoV-2 virus among them.

The second data set of 66,899 SARS-CoV-2 virus sequences was downloaded from the hCoV19 directory of GISAID [23] on 17 July 2020. After data cleaning, all 21 virus sequences with /2017, /2018, or /2019 in their name, presumably incorrectly classified, were excluded. This last set contains three pangolin viruses with NCDs against the selected SARS-CoV-2 virus of, respectively 0.738175, 0.874027, and 0.873367. All of the GISAID data with /2020 in the directory were SARS-CoV-2 viruses. Because one of them had the registration code “hcov-19”, identical to the name of the directory itself, we reduced the data set by removing that sequence: gisaid_hcov-19_2020_07_17_22.fasta.fai hCoV-19 29868 1713295574 80 82. On the sequences initially downloaded from GISAID, we applied a lowercase transformation to each to reduce pointless variability. After that, we computed a histogram of all the characters in the sequence and counted the size of each group. Many sequences contained the base pairs A, C, G, N, T or other letters. We retained the viruses in the list after deduplication and filtering for A, C, G, T. This reduced the GISAID download to a set of unique 15,578 sequences with the known nucleotides A, C, G, and T. Each viral sequence is an RNA sequence and is approximately 30,000 RNA base pairs in size. The total size of all the sequence data together is in the order of two gigabytes.

We then looked at whether there was much variation among the SARS-CoV-2 viruses themselves since this may invalidate the NCD distance between the inspected viruses and the selected SARS-CoV-2 virus. The worst NCD against the selected SARS-CoV-2 virus was 0.874027, namely gisaid hcov-19 2020 07 17 22.fasta<hCoV-19/pangolin/Guangxi/P1E/2017 EPI ISL 410539/2017 from a pangolin. Removing that one sequence from the list, we obtained a worst NCD of 0.873367 as well as from a pangolin in 2017.

Initially, there are 15,430 sequences from GISAID that contain “hCov” in the name. We removed all sequences that contained /2017 in the name. After this, we were left with 15,428 sequences and obtained a worst NCD of 0.738175, also from a pangolin.

After removing the 21 sequences that contained /2017, /2018, or /2019 in the name, partially from pangolins and possibly misclassified according to the 2020 criteria, since the SARS-CoV-2 virus was only established in 2020, 15,409 viruses were left in the list.

To clarify why there is a discrepancy below in that we obtain 15,578 imported sequences but when counting, we lose 100–200 in the next phase: There are some exact name duplicates in the GISAID data. When the “imported sequence” count is reported, then we count identically named identical sequences separately because they did both get imported (but one would overwrite the other). When counting the different names for sequences without /2017, /2018, /2019 in the name, these exact-name duplicates would collapse into a single one, causing just one count.

We retained altogether 66,897 + 6751 = 73,648 raw sequences that were considered in our effort. The GISAID download was reduced after data cleaning to a set of unique 15,578 viral sequences with the known nucleotides A, C, G, and T. This was further reduced to 15,409 sequences. Each viral sequence is an RNA sequence of approximately 30,000 base pairs. The total size of all sequence data together is in the order of two gigabytes. The 6751 unique sequences obtained from the authors of [3] were over the letters A, C, G, and T.

### Selection of a SARS-CoV-2 Virus Isolate as Standard for the NCDs

As a representative of the SARS-CoV-2 viruses, we selected the most common one in the data set as the basis for the NCD against the 6751 non-SARS-CoV-2 viruses. It occurred in the sorted virus list 105 times. In Table 1, it appears at the top with an NCD of 0.00362117 and has the official name of gisaid_hcov-19_2020_07_17_22.fasta<hCoV-19/USA/WI-WSLH-200082/2020|EPI_ISL_471246|2020-04-08. The NCD of the selected SARS-CoV-2 virus against itself should have been 0, but because of the unavoidable compression/computation inaccuracy, it was slightly more.

The worst-case NCD across all the remaining sequences of the SARS-CoV-2 virus to the selected SARS-CoV-2 virus is 0.044986 and the average NCD is 0.009879. The worst-case sequence [24] can be found at gisaid_hcov-19_2020_07_17_22.fasta<hCoV-19/USA/OH-OSUP0019/2020|EPI_ISL_427291|2020-03-31 with registration code EPI_ISL_427291. This shows that the 15,409 SARS-CoV-2 viruses retrieved from GISAID on 17 July in 2020 with /2020 in the name all have sequences that hardly differ from one another.

The results presented in this paper are so much larger than this worst-case NCD that they do not change under subtracting/adding the worst-case NCD to the selected SARS-CoV-2 virus amongst all SARS-CoV-2 viruses. Since the NCD is a metric and thus satisfies the triangle property (Theorem 6.2 [11]), the main results will not be unduly upset by a different choice of selected SARS-CoV-2 virus since the distances involved vary only by at most ±0.044986. Let us illustrate this. Let x,y,z be objects in a metric and let us denote the distance between *x* and *y* by d(x,y), the distance between *x* and *z* by d(x,z), and that between *y* and *z* by d(y,z). Then, |d(x,z)−d(y,z)|≤d(x,y). If *y* is the selected SARS-CoV-2 virus, *x* is an arbitrary SARS-CoV-2 virus, and *z* is one of the non-SARS-CoV-2 viruses, then |d(x,y)|≤0.044986. Thus, the results presented are unaffected by the choice of selected SARS-CoV-2 virus.

## 4. Results

### 4.1. Sorted NCDs

In Table 1, the 60 least NCDs are displayed among all the NCDs between the selected SARS-CoV-2 virus and the 6751 non-SARS-CoV-2 viruses sorted from small to large. The selected SARS-CoV-2 virus appears on the top of the list with the NCD toward itself of 0.00362117. It should be 0, as explained before, and is due to a small error margin in the computation.

The next virus is the only SARS-CoV-2 virus in the database of 6751 virus sequences, but not the selected one, with NCD = 0.0111034. This confirms that the NCDs in the list are accurately calculated since the number is so very small. The code is MN908947.3. It is isolate Wuhan-Hu-1, complete genome GenBank: MN908947.3 of 29903 bp ss-RNA linear VRL 18-MAR-2020 of the family Viruses; Riboviria; Orthornavirae; Pisuviricota; Pisoniviricetes; Nidovirales; Cornidovirineae; Coronaviridae; Orthocoronavirinae; Betacoronavirus; Sarbecovirus.

The following virus has an NCD of 0.444846 with the selected SARS-CoV-2 virus and is the closest apart from the above SARS-CoV-2 virus. It is Sarbecovirus/EPI_ISL_402131.fasta/<BetaCoV/bat/Yunnan/ RaTG13/2013|EPI_ISL_402131. The first part is the classification of the subfamily of viruses; the code EPI_ISL_402131 is that of the virus itself which one can use with Google to obtain further information. The virus isolate was sampled from Rhinolophus affinis, a medium-size Asian bat of the Yunnan Province, China, in 2013. It is 29,855 bp RNA linear VRL 24-MAR-2020, and its final registration code is MN996532. The human coronavirus genome shares at least 96.2% of its identity with its bat relative, while its similarity rate with the human strain of the SARS virus is much lower, only 80.3% [25]. The NCD distance between the selected SARS-CoV-2 virus and this virus is about the same as that between the mtDNA of the chimpanzee and the bonobo according to Appendix A. It is known as bat coronavirus RaTG13 of the family viruses; Riboviria; Orthornavirae; Pisuviricota; Pisoniviricetes; Nidovirales; Cornidovirineae; Coronaviridae; Orthocoronavirinae; Betacoronavirus; Sarbecovirus.

The next three viruses have, respectively, NCD = 0.788416 for Coronaviridae/CoVZC45.fasta/<MG772933.1; NCD = 0.791082 for Coronaviridae/CoVZXC21.fasta/<MG772934.1; and, at a larger distance, NCD = 0.917493 for Coronaviridae/Coronaviridae_783.fasta/<KC881006. Here, the part “fasta/<KC881006” means that the registration code is KC881086.

The first of these three viruses is the Bat SARS-like coronavirus isolate bat-SL-CoVZC45, complete genome at 29,802 bp RNA linear VRL 05-FEB-2020 of the previous family of viruses. Its NCD with the selected SARS-CoV-2 virus is slightly higher than the mtDNA NCD between the human and the gorilla at 0.737 and slightly lower than the mtDNA NCD between the human and the orangutan at 0.834.

The second of these three viruses is the bat SARS-like coronavirus isolate bat-SL-CoVZXC21, with complete genome at 29,732 bp RNA linear VRL 05-FEB-2020 of the same family. The same comparison as that performed previously between the NCD of this virus and that of the selected SARS-CoV-2 virus with the NCDs of mtDNA between species also holds for this second virus.

The third virus of the three is the bat SARS-like coronavirus Rs3367, with a complete genome at 29,792 bp, again of the same family. Comparing the NCD between this virus and the selected SARS-CoV-2 virus yields that it is slightly lower than the NCD between the human mtDNA and the blue whale mtDNA at 0.920 and slightly higher than the NCD between the mtDNA of the finback whale and the mtDNA of the brown bear at 0.915.

In the sense of the NCD, the SARS-CoV-2 virus is therefore likely from the family viruses; Riboviria; Orthornavirae; Pisuviricota; Pisoniviricetes; Nidovirales; Cornidovirineae; Coronaviridae; Orthocoronavirinae; Betacoronavirus; Sarbecovirus.

Figure 1 shows the phylogeny directed binary tree built from the NCD matrix in the Appendix A of the 60 virus sequences in Table 1.

The S(T) values in Figure 1 and Appendix A tell how well the trees represent the distance matrices concerned [11]. To clarify, let there be *n* objects. They have n×n NCDs. Hence, they exist in *n*-dimensional space. Mapping that space onto two dimensions gives distortions of the distances involved. A flat map representing the earth sphere gives such problems of distortion of distances. A tree can represent the n×n distances between *n* objects easier than a more demanding two-dimensional map. The S(T) value tells how well these distances are preserved (0 is not at all and 1 is perfect).

In Figure 1, all sequences are labeled as they occur in the data of [3] together with their registration code. The most interesting are the 11th–15th sequences of the tree from the top of the page. The 13th is EPI_ISL_402131 which is the bat/Yunnan/RaTG13/2013, that is, the RaTG13 bat coronavirus sampled in Yunnan in 2013. The 14th label is the selected SARS-CoV-2 virus which occurs 105 times in the virus database of GISAID, that is, EPI_ISL_428253. The 15th label is MN908947, which is the SARS-CoV-2 virus Wuhan Hu-1 from the Wuhan seafood market collected in December 2019, submitted 05 January 2020, and reported in *Nature*, 579(7798), 265–269, 270–273 (2020). This is the only SARS-CoV-2 virus in the 6751 sequences obtained from the authors of [3]. The 11th and 12th labels are the CoVZC45 bat coronavirus and the CoVZXC21 bat coronavirus. Numbers 11, 12, 13, 14, and 15 have the least NCD distances to the selected SARS-CoV-2 virus. The entire 60×60 NCD matrix underlying this tree is too large to display here but is supplied in the Appendix A.

### 4.2. Thirty-Seven Viruses Close to the Selected SARS-CoV-2 Virus

Figure 2 contains the selected SARS-CoV-2 sequence and all the GISAID sequences ending in /2017, /2018, and /2019 which include the three pangolin viruses in the second paragraph of Section 3 on Materials or the Data Cleaning section of the Appendix A. A dozen close sequences from all GISAID sequences and a dozen close sequences from the machine learning approach study [3] data were added. The ladders in the directed binary tree are usually there to accommodate more than two outgoing branches from a single node. The entire 37×37 NCD matrix is too large to display here but is supplied in the Appendix A.

## 5. The Pangolin Connection

As we saw in the section on data cleaning in the Appendix A, the GISAID hCoV19 database is littered with pangolin viruses (now called pangolin-CoV) from 2017, 2018, and 2019. Several studies, e.g., [26,27], hold that while the SARS-CoV-2 virus probably originates from bats, it may have been transmitted to another animal and/or recombined with a virus there and zoonotically transmitted to humans. The other animal is most often identified as the pangolin. The compression method shows that the NCDs between the pangolin-related virus and the human SARS-CoV-2 virus are far apart. However, they are not farther apart than 0.738175–0.874027. This is equivalent to the NCD distance between the mtDNA of a human and a gorilla up between the mtDNA of a gray seal and a blue whale. The bat-SL-CoVZXC21 and bat-SL-CoVZC45 viruses are at NCD = 0.791082 and NCD = 0.788416. This is in between the mtDNA NCD distance of a human versus gorilla at 0.737 and the mtDNA distance of a human versus an orangutan at 0.834.

However, the RaTG13 virus has an NCD = 0.444846 distance from the selected SARS-CoV-2 virus which is close to one half of the pangolin NCDs given here. As noted before, this is comparable with the NCD between the chimpanzee and the bonobo according to Appendix A. Furthermore, in the tree of Figure 2, the pangolin viruses are generally far from the selected SARS-CoV-2 virus.

## 6. Discussion

We determined the phylogeny and taxonomy of the SARS-CoV-2 virus. Earlier studies using alignment-based methods have suggested that the SARS-CoV-2 virus originated from bats. Bats are a known reservoir of viruses that can zoonotically transmit to humans [28]. The machine-learning approach, an alignment-free method [3], came to the same conclusion. The current, completely different, approach agrees with this and has as an intermediary candidate—the pangolin.

The method used herein [11,13] is based on compression, alignment-free, remarkably simple, very fast, and quantifies a distance in a number called the NCD between 0 and 1, where 0 is identical and 1 is totally different. This uses only the viral DNA/RNA sequences. It is briefly explained in Section 7 and illustrated by the NCDs of the mtDNA of 24 mammal species and the phylogeny of the SARS virus in the Appendix A. In practice, the compression method performs accurately [11,13,15,16,18].

To compare the compression method with the alignment-free method based on machine learning [3], we used the latter database of over 6500 unique viral sequences. To select the particular SARS-CoV-2 virus against which these viral sequences are compared, we used a database of approximately 15,500 unique SARS-CoV-2 viruses. The NCD analysis places the RaTG13 bat virus closest to the SARS-CoV-2 virus followed by the two SARS-like coronaviruses bat-SL-CoVZXC21 and bat-SL-CoVZC45 just like the machine learning method and alignment-based methods. Additionally, some pangolin viruses are close and over 6500 viruses identified by their registration code in the database of [3] are farther away (the codes of slightly less than 60 of them are given in Table 1).

For alignment-based methods on the viruses of the database of [3], such a feat is complex or infeasible. On the other hand, these methods yield biological interpretations. The alignment-free machine-learning method [3] determines the taxonomy of a single virus sequence and gives different phylogeny trees based on different aspects of it. The compression method gives a single phylogenetic tree for a set of many DNA/RNA sequences. The compression method is domain-independent and requires no parameters to be set, apart from the used compression algorithm. Obtaining essentially the same main positive results as the earlier studies, and agreeing with the generally believed hypothesis, this method is less complicated than previous methods. It yields quantitative evidence that can be compared with the NCDs among the mtDNA of familiar mammals. Since the method is uncomplicated and very fast, it is useful as an exploratory investigation into the phylogeny and taxonomy of viruses of new epidemic outbreaks.

## 7. Mathematical Derivation of the NCD

In 1936, A.M. Turing [29] defined the hypothetical “Turing machine”, whose computations give the universally accepted formal definition of the intuitive notion of computability in the discrete domain. These Turing machines compute integer functions, the *computable* functions. By using pairs of integers for the arguments and values, we can extend computable functions to functions with rational arguments and/or values.

### 7.1. Kolmogorov Complexity

Informally, the Kolmogorov complexity of a string is the length of a shortest binary string from which the original string can be reconstructed by a general-purpose computer such as a particular universal Turing machine *U*. Hence, it constitutes a lower bound on how far a lossless compression program can compress. For details, see the text [30]. In this paper, we require that the set of programs of *U* is prefix free, which means that no program is a proper prefix of another program, that is, we deal with the *prefix Kolmogorov complexity*. We call *U* the *reference universal prefix Turing machine*. Formally, the *conditional prefix Kolmogorov complexity* K(x|y) is the length of a shortest binary input *z* such that the reference universal prefix Turing machine *U* on input *z* with auxiliary information *y* outputs *x*. The *unconditional prefix Kolmogorov complexity*
K(x) is defined by K(x|ϵ), where ϵ is the empty word of length 0. The functions K(·) and K(·|·), though defined in terms of a particular machine model, are machine-independent up to an additive constant and acquire an asymptotically universal and absolute character through Church’s thesis, and from the ability of universal machines to simulate one another and execute any effective process.

The Kolmogorov complexity of an *individual finite object* was introduced by Kolmogorov [31] as an absolute and objective quantification of the amount of information in it. It is sometimes confused with the information theory of Shannon [32], which deals with *average* information *to communicate* objects produced by a *random source*. They are quite different.

### 7.2. Information Distance

The *information distance* D(x,y) between strings *x* and *y* is defined as
$$D\left(x,y\right)=\min_{p}\left\{|p|:U\left(p,x\right)=y\wedge U\left(p,y\right)=x\right\},$$
where *U* is the aforementioned reference universal prefix Turing machine. As in the case of the Kolmogorov complexity *K*, the distance function *D* is upper semicomputable. ‘Upper semicomputable’ means computable from above in the limit, but not necessarily from below. Define:E(x,y)=max{K(x|y),K(y|x)}. (here and elsewhere in this paper, “logarithm” or “log” refers to the binary logarithm). In [12], it is shown that the function *E* is upper semicomputable, D(x,y)=E(x,y)+O(logE(x,y)), the function *E* is a metric (more precisely, it satisfies the metric (in)equalities up to a constant), and *E* is minimal (up to a constant) among all upper semicomputable distance functions $D^{\prime }$ satisfying the normalization conditions $\sum_{y:y\neq x}2^{-D^{\prime }\left(x,y\right)}\leq 1$ and $\sum_{x:x\neq y}2^{-D^{\prime }\left(x,y\right)}\leq 1$ (to exclude bogus distances which state, for example, that every *y* is at $\frac{1}{2}$ distance of a given *x*). We can show that this metric *E* is *universal* [12] in the sense that for every pair of finite files, x,y, we have that E(x,y) is at least as small as the smallest $D^{\prime }\left(x,y\right)$. This means that E(x,y) is at least as small as the distance engendered by the dominant feature shared between *x* and *y*. The *normalized information distance* (NID) is defined by
(2)$$NID\left(x,y\right)=\frac{E(x,y)}{\max\{K(x),K(y)\}}.$$ It is straightforward that 0≤NID(x,y)≤1 (up to an O(1)/max{K(x),K(y)} additive term). It is a metric [13] (and so is the NCD in (Equation 1), see [11]). Since by the symmetry of information law [33] or see [30]:(3)$$\begin{array}{ccc} K(x,y) & = & K(x)+K(y|x)+O(\log(K(x)+K(y)))\\ & = & K(y)+K(x|y)+O(\log(K(x)+K(y))), \end{array}$$
rewriting the NID using Equation 3 yields:(4)$$NID\left(x,y\right)=\frac{K(x,y)-\min\{K(x),K(y)\}}{\max\{K(x),K(y)\}},$$
up to some small additive terms that we ignore. For more details on this derivation, see [30] or [34]. (As an aside, a nonoptimal precursor to the NID/NCD was given in [35]).

In this way, in [12,13], we and others developed theoretical approaches to the similarity of finite objects. We proved that these theories based on the Kolmogorov complexity are perfect.

### 7.3. Normalized Compression Distance

Unfortunately, the universality of the NID comes at the price of incomputability. In fact, the NID is not even semicomputable and there is no semicomputable function at a computable distance of it [36].

However, the length of the compressed version of a finite string, such as a compressed computer file, is obviously computable. For the natural data we are dealing with, practice evidences [37] that the length of the binary representation of the compressed version of a file seems not too far from its Kolmogorov complexity.

Therefore, if *Z* is a compressor, Z(x) denotes the length of the binary compressed version of a string *x*, substituting *Z* everywhere in Equation (Equation 4) by the prefix Kolmogorov complexity *K*, we obtain Equation (Equation 1). Hence, by approximating the Kolmogorov complexities involved through real-world compressors, we transformed these theoretical notions into practical ones and can use them to establish similarity in natural data [11,38].

## Figures and Tables

**Figure 1 entropy-24-00439-f001:**
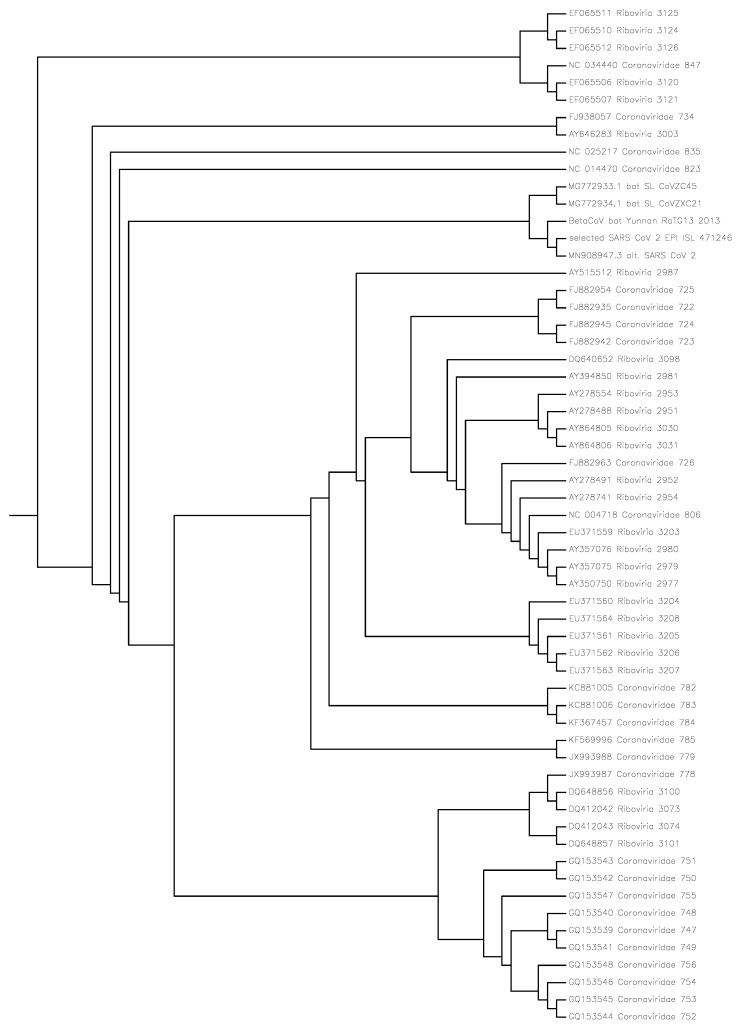
These 60 viruses have the least NCD with the SARS-CoV-2 virus. The figure therefore gives a visual representation of the structural relations close to the SARS-CoV-2 virus. We use human mtDNA as an outgroup; it is approximately the same size as the viruses and it can be assumed to be completely different. Using an unrelated sequence which is called the “outgroup” is a common method to determine where the root of a phylogenetic tree is: where it joins the tree, there is the root. Since S(T)=0.998948, the tree almost perfectly represents the NCD distance matrix concerned in the Appendix A.

**Figure 2 entropy-24-00439-f002:**
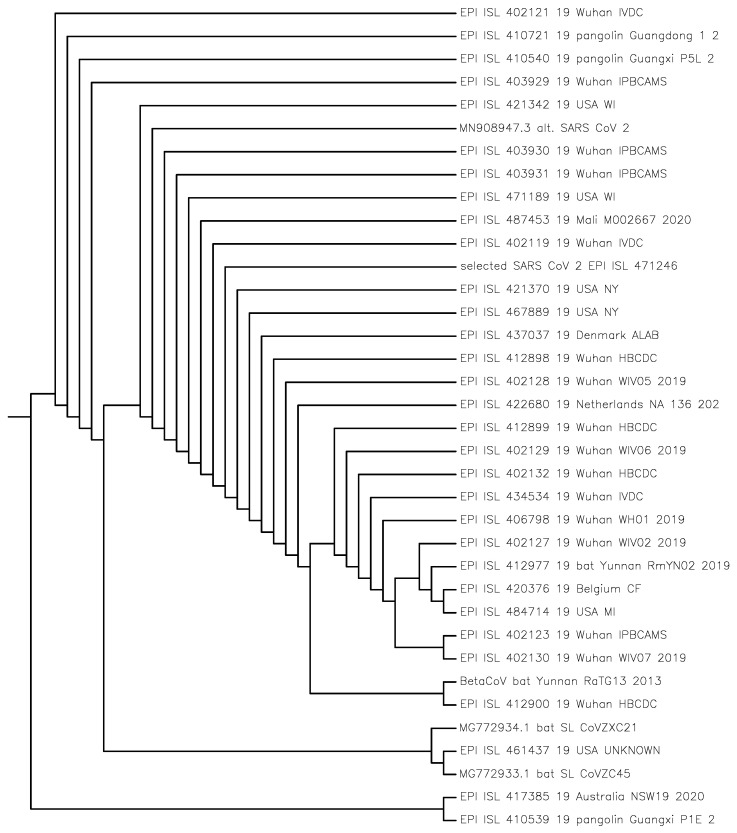
The phylogenetic directed binary tree built from 37 virus sequences with the human mtDNA as an outgroup to determine the root.

**Table 1 entropy-24-00439-t001:** First 60 items in the sorted list of the 6751 virus sequences used.

NCD	Virus Name
0.00362117	selected_SARS_CoV_2_EPI_ISL_471246
0.0111034	MN908947.3_alt._SARS_CoV_2
0.44486	BetaCoV/bat/Yunnan/RaTG13/2013|EPI_ISL_402131_EPI_ISL_402131
0.788416	MG772933.1_bat_SL_CoVZC45
0.791082	MG772934.1_bat_SL_CoVZXC21
0.917493	KF569996_Coronaviridae_785
0.917563	KC881006_Coronaviridae_783
0.91801	KC881005_Coronaviridae_782
0.918257	FJ882963_Coronaviridae_726
0.918381	AY278554_Riboviria_2953
0.918447	EU371561_Riboviria_3205
0.918497	AY278488_Riboviria_2951
0.918531	AY278741_Riboviria_2954
0.918553	AY278491_Riboviria_2952
0.918565	NC_004718_Coronaviridae_806
0.918597	EU371563_Riboviria_3207
0.918605	FJ882935_Coronaviridae_722
0.918658	AY357075_Riboviria_2979
0.918669	EU371559_Riboviria_3203
0.918691	DQ640652_Riboviria_3098
0.918724	EU371562_Riboviria_3206
0.918796	AY350750_Riboviria_2977
0.918829	AY864805_Riboviria_3030
0.918945	FJ882945_Coronaviridae_724
0.919072	EU371560_Riboviria_3204
0.919117	AY864806_Riboviria_3031
0.919182	EU371564_Riboviria_3208
0.919221	AY394850_Riboviria_2981
0.919244	FJ882942_Coronaviridae_723
0.919486	AY357076_Riboviria_2980
0.91954	FJ882954_Coronaviridae_725
0.91993	KF367457_Coronaviridae_784
0.920486	AY515512_Riboviria_2987
0.921053	JX993988_Coronaviridae_779
0.923045	GQ153543_Coronaviridae_751
0.923151	GQ153542_Coronaviridae_750
0.92541	DQ648857_Riboviria_3101
0.925706	GQ153547_Coronaviridae_755
0.925802	JX993987_Coronaviridae_778
0.925844	GQ153544_Coronaviridae_752
0.925951	GQ153548_Coronaviridae_756
0.925982	GQ153545_Coronaviridae_753
0.926013	GQ153540_Coronaviridae_748
0.92613	GQ153546_Coronaviridae_754
0.92615	GQ153539_Coronaviridae_747
0.92615	GQ153541_Coronaviridae_749
0.926681	DQ412043_Riboviria_3074
0.931577	DQ412042_Riboviria_3073
0.932533	DQ648856_Riboviria_3100
0.952228	NC_014470_Coronaviridae_823
0.994546	NC_025217_Coronaviridae_835
0.994897	NC_034440_Coronaviridae_847
0.994986	FJ938057_Coronaviridae_734
0.994986	AY646283_Riboviria_3003
0.995078	EF065512_Riboviria_3126
0.995078	EF065511_Riboviria_3125
0.995078	EF065510_Riboviria_3124
0.995086	EF065506_Riboviria_3120
0.995086	EF065507_Riboviria_3121
0.995086	EF065505_Riboviria_3119

## Data Availability

The analytic software which was used is available at GitHub source code URI: https://github.com/rudi-cilibrasi/ncd-covid accessed on 16 September 2020. We downloaded the data broadly in two parts, stored them on the public repository GitHub and linked to it here: https://github.com/rudi-cilibrasi/ncd-covid#downloading-input-data accessed on 16 September 2020. We also clarified the instructions for downloading the GISAID data in the same page and provided a link. Users will have to register a free account with GISAID since they have a strict “no redistribution” policy.

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
