# Peer review of "Fast Phylogeny of SARS-CoV-2 by Compression"

_entropy, 2022, doi:10.3390/e24040439_

Round 1

Reviewer 1 Report

Summary: The paper applies the well-known NCD method to compare COVID-19 virus SARS-CoV-2 to 6500 other viruses. The resulting taxonomy closely matches other methods.
The paper applies a known method to a known data set to confirm known results. As such, the paper has little scientific merit. Nevertheless I think it should be published. The importance of this paper stems from the fact that: * it tackles a very timely and extremely important problem * After nearly 20 year since the introduction of the NCD method, it has still not become a default method in the toolbox of geneticists, which is a gross oversight, especially in light of problems of global importance such as COVID-19.
I could not find the Beta, Gamma, Delta, and Omicron variants of the COVID-19 virus SARS-CoV-2 analysed in the paper. Maybe they are somewhere hidden with some cryptic names. In any case, these variants should be included and some attention paid to them, e.g. how close they are to each other and relative to the other virus.
I'm in favor of publishing the paper if the authors discuss the popular COVID-19 virus SARS-CoV-2 variants (Beta, Gamma, Delta, Omicron, ...)
Further comments:
* A lot of irrelevant details were described in the paper, including the methods by which the data was cleaned by removing particular dates, or by listing long names of particular sequences that were removed. A nicer approach would be to explain in general terms how the data was cleaned, and include a link to a repo of the cleaned data, together with more details of precisely what was done to the data. To include this in the main paper is rather distracting. Is it relevant to the reader to list out the full name of Wuhan-Hu-1 and other such diseases?
* It's not clear that comparing NCDs between viruses and human/chimpanzees for a sense of closeness makes sense. For all the reader knows, all animals in class mammalia have as high a NCD between them as that for humans and chimpanzees. Perhaps a table of the NCD's between a human and other animals might be useful as a reference for how strong the similarity is implied by NCD=0.79...
* The same goes for the comparison of the NCD with that of the Finback Whale/Brown bear. Should it be surprisingly that their genome sequences have a NCD of 0.915? Are most animals this dissimilar, and so a NCD of ~0.9 for viruses may not be that surprising?
* The distortion measure S(T), while covered in the referenced paper, could be (at least informally) described in the paper provided, enough to at least provide some intuition.
* Not clear why we use an outgroup to determine where the root of the tree is.
* The tree should highlight clearly where the SARS-CoV-2 virus is, or have another copy of the tree included that points the reader onto the section of the tree containing the viruses most similar to SARS-CoV-2. Perhaps the leaves of the tree could also include the NCD values?
* I don't think it's useful to include several pages of the 37 x 37 NCD matrix, but could instead be hosted on a repo with the data used for this experiment, and represented in a graphical way (a 37x37 heat map, for instance) in the paper.
* Not clear how strong a conclusion can be drawn from a given NCD value. How close to 1 does the NCD need to be before we can say that two viruses are similar/can draw a conclusion that SARS-CoV-2 came from a particular animal?
* Does it makes sense to say that one NCD is close to one-half of another? Is that something that makes sense to do with an NCD value?

Reviewer 2 Report

The article entitled “Fast Phylogeny of SARS-CoV-2 by Compression” present interesting approach and the work is designed properly, however, how accurate it is this method. Do you have additional proof that it is takes in consideration in present study? What about selectivity and specificity of the method?  

Author Response

In the report of Referee #1 he/she writes:

The article entitled “Fast Phylogeny of SARS-CoV-2 by Compression” present interesting approach and the work is designed properly, however, how accurate it is this method. Do you have additional proof that it is takes in consideration in present study? What about selectivity and specificity of the method?  

Answer:

The mathematical derivation is given in Section 7 and especially Section 7.3, while the practical use of it id validated in the section "Validation of the method":

the earlier version of NCD, USM has also been used  for protein structure comparison [1,2] and and NCD has even been used to automatically -and blindly- classify optimisation problem instances[3], thus I think the validity of NCD as a general -very practical- tool is well established.

[1] Measuring the similarity of protein structures by means of the universal similarity metric. Bioinformatics. 2004 May 1;20(7):1015-21. doi: 10.1093/bioinformatics/bth031. Epub 2004 Jan 29. PMID: 14751983.

[2] ProCKSI: a decision support system for Protein (structure) Comparison, Knowledge, Similarity and Information. BMC Bioinformatics. 2007 Oct 26;8:416. doi: 10.1186/1471-2105-8-416. PMID: 17963510; PMCID: PMC2222653.

[3] Blind optimisation problem instance classification via enhanced universal similarity metric. Memetic Comp. 6, 263–276 (2014). https://doi.org/10.1007/s12293-014-0145-7

--------------------

Reviewer 3 Report

Fast Phylogeny of SARS-CoV-2 by Compression

The authors proposed a compression-based distance method to perform phylogenetic analysis. In a nutshell, the compressed distance between a taxon (sample) and a reference (e.g., SARS-CoV-2) serves as a proxy for evolutionary divergence. They claimed that their method was faster, and the results closely matched other methods, alignment-based and machine-learning-based methods. Overall, the article did not substantiate its claims with thorough comparisons.

Major comments:

  1. They claimed their method was faster than existing methods. Surprisingly, there is not a single table stating what methods they considered, testing datasets, and performance statistics.
  2. Importantly, runtime performance is only one of the many criteria in accessing computational phylogenetic methods. For instance, the capability to identify conserved nucleotides or amino acids can inform vital biological functions. But such capability is totally lacking in this method.
  3. Their method is distance-based, i.e., computing similarity or dissimilarity between two sequences as a whole like NJ tree. However, they did not compare with the other school of phylogenetic methods, which are site-based, e.g., maximum likelihood tree. How does their method fair with ML tree?
  4. #12, "We treat the question whether Pangolins are involved in the SARS-CoV-2 virus." I am confused about how their method can tackle this question.
  5. #66. The trees in Figures 1 and 2 were created by PHYLIP, but they did not describe the parameters and distance methods used. But in the figure legend (around #260), it said NCD was used. So, I am confused about how they created the two trees.
  6. #72. Why is the affiliation section here?
  7. #115. They chose zpaq compression method but didn't state what NCDzpaq(x,x) was.
  8. They describe the intuition of NCD in the Method section (from #77). However, many important questions remain unanswered. They should calibrate NCD by the percentage of mismatches between two sequences. E.g., randomly mutated 5%, 10%, 15%, etc. of the reference sequence. Does the NCD change proportionally? Additionally, what is the NCD between sequences x and x', where x' is a substring of x?
  9. #227-229. Unsure of the rationale of incorporating primate mt sequences.
  10. #135. The whole section should belong to 3. Materials.
  11. #245. What is S(T) value?
  12. There are many erroneous citations.

Round 2

Reviewer 1 Report

The authors rebuttal is unsatisfactory. As I wrote in my first review "The paper applies a known method to a known data set to confirm known results. As such, the paper has little scientific merit. On the other hand" I am willing to overlook all this if and only if the paper is a (1) "hot-topic" paper *and* (2) written with biologist readers in mind.
to 1) I find the response that the experiments cannot be diluted, and that the standard Omicron variant cannot be extracted unconvincing. In the worst case the authors could simply sub-sample some Omicron variants (and same for Beta,Gamma,Delta) .
to 2) Regarding "The S(T) value is introduced and described in reference [8] and the intuition is clear: close to 1 means (almost) no distortion and less than say 0.8 means a lot of distortion. For in tuition the reader should read [8]."
I have no idea what "a lot of distortion" looks like, and would want at least some sort of explanation or examples to demonstrate the scale of S(T) from 0 to 1. [8] describes S(T), but also gives no intuition.
What does an NCD of 0.9 look like in practice? Including intuition/scales for S(T) and NCD are all the more important given that part of the audience of this paper is people in the field of biology who have likely never heard of NCD. Presumably we want NCD to be a more commonly used method, and if I were a biologist, this paper would not encourage me to use it.
I think it would be possible to give meaning to the absolute values of the NCD, and to some extent necessary for this paper to be interesting to biologists:
For instance, one could choose some appropriate data set S (like genome sequences of animals) and pick a particular element X of that data set (say, human), and measured the distance between X and all the elements of S. Then, add an image of a number line from 0 to 1, and include on the line all the distances from an element of S to X, labelled as to what they are. This was done in the paper in a very weak fashion (comparing human to chimpanzee) but I think it'd be useful to include a large collection of animals to get a sense of scale of NCD (saying NCD(human, chimp) = 0.76 isn't useful if it were the case that NCD(fish, human) was also about 0.76, for instance.
There are also no comparisons of computational load of the CND method v.s the typical methods used in the field of medicine, though I understand this data may be difficult to obtain.

Author Response

Rebuttal Reviewer 1.

The answer of reviewer 1 raises interesting points. However, there are various reasons not to pursue these avenues.

Ad 1) For the following reasons this is not a good suggestion and cannot be followed.

a) The GISAID data base contains over 1.5 million omicron isolates. It is infeasible to compute a representative so that the NCD distance to the selected SARS-Covid-2 isolate can be established. Choosing just 1 or a small number of omicron isolates from this mulitude may be meaningless since it could well be that another choice would have vastly different results.

b) Since in omicron there are about 50 mutations on about 30,000 base pairs it is to be expected that the compression method cannot distinguish these.

c) The work involved whatever way we do it is another paper and the results with the method may be nihil.

d) The proposal amounts to an anachronism w.r.t. the rest of the paper which gives the situation on 17-07-2020 and destroys the unity of the approach.

Ad 2) The S(T) value and its computation is extensively discussed in reference [8] as referred to together with examples and what it means. It is not desirable and permitted to repeat this. If one is interested consult [8] which is readily available.

Reviewer 2 Report

Thank you for the answert

Author Response

Thank you for asking no further questions

Reviewer 3 Report

I saw the runtime results in #60-65. But what is the baseline for comparison? "Fast" is subjective.

I stand with my original comments 2, 7, and 8. 
